# OpenReview forum: "Tensor Trust: Interpretable Prompt Injection Attacks from an Online Game"
_ICLR.cc/2024/Conference — ICLR 2024 spotlight_

### Official Review · Reviewer_6PUw · 2023-11-02

**Soundness:** 3 good
**Presentation:** 3 good
**Contribution:** 2 fair
**Rating:** 8
**Confidence:** 2

**Summary:**

The paper introduces a dataset designed to evaluate the robustness of Large Language Models (LLMs). The dataset is collected using a carefully crafted game called Trust Tensor. Through this game, over 100,000 attack prompts and 46,000 defence prompts are gathered. By examining the attack and defence pairs, the study reveals that successful attacks often exhibit a simple structure, highlighting the adversarial vulnerabilities of LLMs.

**Strengths:**

The game Trust Tensor is well-designed and has the potential to serve as a benchmark for evaluating the adversarial robustness of Large Language Models.

**Weaknesses:**

The task of prompt extraction is distinct from the Trust Tensor. In other words, the Trust Tensor is not very suitable for collecting the prompt extraction dataset.

**Questions:**

See Weaknesses.

---

> ### Author Response · Authors · 2023-11-17
> **Response to Reviewer 6PUw**
>
> Thank you for your review. We’re glad to hear you found our game to be well-designed and useful as an adversarial robustness benchmark! We've uploaded a new revision of the paper with changes summarized in the top-level shared response. We'd also like to offer a clarification on prompt extraction:
>
> > The task of prompt extraction is distinct from the Trust Tensor. In other words, the Trust Tensor is not very suitable for collecting the prompt extraction dataset.
>
> You're right that the game does not explicitly reward players for performing prompt extraction, but many players found that an effective way to achieve "access granted" was to first perform prompt extraction, read the password from the prompt, and then enter the password verbatim. See the middle row of Figure 1 for an example of this. Around 27% of successful attacks on other players contain the defender's exact access code (indicating that the attacker performed successful prompt extraction), and the prompt extraction robustness benchmark contains a subset of ​​569 strong, human-verified examples of prompt extraction across different defenses. (The number 569 has increased from the original submission, as we expanded and manually reviewed our benchmark subsets for the latest revision.)

---

### Official Review · Reviewer_ZAuw · 2023-11-03

**Soundness:** 3 good
**Presentation:** 3 good
**Contribution:** 2 fair
**Rating:** 5
**Confidence:** 3

**Summary:**

The authors present a large dataset of human-generated adversarial examples for instruction-following LLMs. The defense and attack strategies are identified by different methods, and some insights on attacks and defense are given. This proposed dataset is also used to evaluate the robustness of LLMs.

**Strengths:**

[1]	The proposed dataset has been released publicly.

[2]	The samples in the dataset are high-quality since they are devised manually.

**Weaknesses:**

[1]	Although the size of the dataset is pretty large, the mechanism of this game is monotonous.

[2]	I am unsure whether the topic of this paper aligns with the theme of this conference.

**Questions:**

[1]	What are the potential applications of this dataset? In other words, how can it help improve the robustness of LLMs?

[2]	I wonder whether the attack and defense in the dataset are still effective if the prompt of the opening defense is changed.

---

> ### Author Response · Authors · 2023-11-17
> **Response to Reviewer ZAuw**
>
> Thank you for your review and suggestions for improvement. We’ll address these points one by one below. We've also uploaded a new revision of the paper with various general improvements (see shared top-level response). We hope we have addressed all your concerns; please consider revising your score if this is the case.
>
> ## Weakness 1: Game mechanism
>
> > [1] Although the size of the dataset is pretty large, the mechanism of this game is monotonous.
>
> We're not sure whether "monotonous" in this context means "not enjoyable" or "generates uninteresting data", so we'll discuss both:
>
> * **Do players like it?** Tensor Trust is a simple game, but players appear to enjoy it. After our initial promotion push, much of our web traffic has been driven by organic sharing on social media (i.e. not due to our own promotion), which continues to drive a few thousand attacks per day. As of November 16th, we have >3,000 registered accounts and >330,000 submitted attacks, as well as >250 players on our Discord, which shows strong engagement.
> * **Is the data interesting?** We chose a simple objective for the game (make the LLM say "access granted") so that it would be easy to objectively evaluate attack success (as opposed to, e.g., jailbreaking attacks, which require subjective evaluation). Despite its simplicity, we've found that this objective is sufficient to elicit a rich dataset of attacker behaviors, including prompt extraction as a means to gain additional information (as opposed to just prompt hijacking), and the various classes of attack outlined in Table 1. Overall, we found that the game setup strikes a good balance between making it easy to do objective evaluation of attack/defense strategies and encouraging diverse techniques.
>
> ## Weakness 2: Relevance to ICLR
>
> > [2] I am unsure whether the topic of this paper aligns with the theme of this conference.
>
> We think that ICLR would be a good venue for this paper due to its history of publishing strong benchmark datasets and adversarial robustness work. In the past, ICLR has been a venue for seminal work in this area, such as the [ImageNet-C dataset for robust computer vision](https://openreview.net/forum?id=HJz6tiCqYm) (ICLR'19, now cited 2.6k times), or [this popular vision-based adversarial example defense](https://openreview.net/forum?id=rJzIBfZAb#) (ICLR'18, now cited >10k times). More recently, ICLR 2023 published a variety of papers that consider datasets/benchmarks (examples: [[1]](https://iclr.cc/virtual/2023/poster/11601), [[2]](https://iclr.cc/virtual/2023/poster/11422), [[3]](https://iclr.cc/virtual/2023/poster/11900)) or adversarial robustness (examples: [[1]](https://iclr.cc/virtual/2023/poster/11455), [[2]](https://iclr.cc/virtual/2023/poster/11946), [[3]](https://iclr.cc/virtual/2023/poster/11270)). It also had two workshops on robustness/security/privacy ([[1]](https://iclr.cc/virtual/2023/workshop/12827), [[2]](https://iclr.cc/virtual/2023/workshop/12825)). Both "societal considerations including fairness, safety, privacy" and "datasets and benchmarks" are listed as relevant topics in [this year's CFP](https://iclr.cc/Conferences/2024/CallForPapers), so we expect this kind of work will continue to be of interest to the broader community at ICLR 2024.
>
> ## Question 1: Applications
>
> > [1] What are the potential applications of this dataset? In other words, how can it help improve the robustness of LLMs?
>
> This is a good point, and we've modified the paper to make this more clear. To summarize, here are some ways in which the data and experiments in the paper can help improve LLM robustness:
>
> * Researchers can use our robustness benchmarks (section 3/section 6) to measure whether newly released LLMs (or new forms of LLM finetuning) improve robustness to prompt hijacking and prompt extraction.
> * Our taxonomy of strategies (section 4) and qualitative analysis in the appendix shows various ways in which existing LLMs fail to be robust to prompt injection, which we believe is useful to target future efforts in fine-tuning LLMs against prompt injection (e.g. the analysis in appendix H shows that system-user message roles have little effect on GPT 3.5 Turbo, which highlights the need for stronger steerability for these types of models). This qualitative analysis may also be useful for prompt engineers and red-teaming efforts.
> * We also hope that our data will be useful for directly training better-defended models (by optimizing them to ignore the adversarial examples in our dataset), as well as for automated red-teaming (e.g. learning multi-step attack strategies from the data). We leave these more complex applications to future work.
>
> We've  now improved the contribution statement in the introduction to make the first two contributions (evaluation benchmark, interesting qualitative analysis) more clear, and highlighted the potential future work in the third point in the conclusion.
>
> (continued in reply)

---

> > ### Author Response · Authors · 2023-11-17
> > **Continuation of response**
> >
> > (continued)
> >
> > ## Question 2: Changing the opening defense
> >
> > > [2] I wonder whether the attack and defense in the dataset are still effective if the prompt of the opening defense is changed.
> >
> > It depends on the structure of the defense. In some defenses the opening defense matters quite a bit; e.g. this might be because the defense only mentions the password in the opening defense, had a very short closing defense, or even chose to leave the closing defense blank. Removing the opening defense from these defenses would have rendered them ineffective against attackers, or would have made the access code invalid.
> >
> > Our dataset could be useful to researchers trying to answer questions about the relative value of opening defenses versus closing defenses. For instance, you could construct paired defenses with different opening defenses but the same closing defense and investigate how much robustness rate changes our hijacking and extraction benchmarks. We leave this kind of analysis to future work on novel defense techniques.

---

> ### Author Response · Authors · 2023-11-21
> **Response to Reviewer ZAuw**
>
> Thank you again Reviewer ZAuw for your thoughtful feedback. We have attempted to address your concerns in our rebuttal and in our latest revision of the paper. We would appreciate it if you could look over our rebuttal and update your score if it has addressed your concerns. We're also happy to address any additional concerns you might have before the end of the discussion period on November 22 (tomorrow).

---

### Official Review · Reviewer_gj2Z · 2023-11-09

**Soundness:** 3 good
**Presentation:** 3 good
**Contribution:** 3 good
**Rating:** 8
**Confidence:** 3

**Summary:**

This paper introduces a novel dataset of over 100,000 prompt injection attacks and 46,000 defense prompts, generated by players in an online game called Tensor Trust. These attacks target Large Language Models (LLMs) and shed light on the vulnerabilities of LLM-based applications to malicious prompt manipulations. The dataset is the largest of its kind, and it reveals that LLMs are susceptible to both prompt extraction and prompt hijacking attacks. The authors use this dataset to create benchmarks to evaluate LLM resistance to such attacks and demonstrate that many models are vulnerable. They also show that these attack strategies generalize to real-world LLM applications beyond the game setting.

**Strengths:**

* The use of an online game to collect human-generated adversarial examples for instruction-following LLMs is a novel and creative way to understand the weaknesses of these models.

* The creation of benchmarks for evaluating LLM resistance to prompt injection attacks is a valuable contribution, as it provides a standardized way to assess the security of these models. This is of practical importance in the development of secure LLM-based applications.

**Weaknesses:**

* The paper's setting is initially hard to grasp; authors should aim to explain the threat model clearly, using notations and specifying the level of access attackers have.

* The paper should establish a connection to Textual Backdoor attacks, even though these attacks typically require a more significant level of access to LLMs or their pretraining data than the setting the authors are primarily interested in. This additional context would help improve the clear understanding of the threat model in which Tensor Trust operates.

**Questions:**

See above

---

> ### Author Response · Authors · 2023-11-17
> **Response to Reviewer gj2Z**
>
> Thank you for your review! We're glad that you agree that benchmarking resistance to prompt injection is a valuable contribution and that you found our work novel. We also found your feedback in the weaknesses section particularly helpful, and have updated the paper in response. We've uploaded a new copy of the paper (please see the shared top-level message for details), and we address your specific points in more detail below.
>
> ## Weakness 1
>
> > The paper's setting is initially hard to grasp; authors should aim to explain the threat model clearly, using notations and specifying the level of access attackers have.
>
> Our new revision uses more precise notation in the introduction, and additional detail on the attacker and defenders' abilities and goals. The relevant paragraphs now read as follows:
>
> > The Tensor Trust web game simulates a bank. Each player has a balance, which they can increase by either coming up with successful attacks or by creating a defense that rebuffs attacks. In this section, we will explain the mechanics of defending and attacking, as well as implementation details on how we evaluate the attacks and defenses submitted by users.
> >
> > **Notation**
> > We use $\mathcal V$ to denote a token vocabulary and $L : \mathcal V^* \times \mathcal V^* \times \mathcal V^* \to \mathcal{V}^*$ to denote an LLM that takes in three strings and outputs a single response string. $G : \mathcal V^* \to \{T, F\}$ denotes a *goal predicate* that determines whether a string says  "access granted" (achieved using the regex in Appendix B).
> >
> > **Defending**
> > Each account has a _defense_ which consists of three prompts: an opening defense $d_{\text{open}}$, an access code $c_{\text{access}}$, and a closing defense $d_{\text{close}}$, as shown in Fig. 2. When a user saves a defense, we validate it by sandwiching their access code between the opening and closing defense and feeding it to the LLM $L$. The access code can only be saved if it makes the LLM output "access granted". In other words, $G(L(d_{\text{open}}, c_{\text{access}}, d_{\text{close}}))$ must be true.
> >
> > **Attacking**
> > A player can select any other player's account and submit an attack against it. The text of the first player's attack, $c_{\text{attack}}$, is sandwiched between the defending player's opening and closing defense ($d_{\text{open}}$ and $d_{\text{close}}$), and then fed into the LLM $L$. If the LLM outputs "access granted" (i.e., $G(L(d_{\text{open}}, c_{\text{attack}}, d_{\text{close}}))$ is true), the attacker steals a fraction of the defender's money. Otherwise, the defender is granted a small amount of money for rebuffing the attack. The attacker cannot see $d_{\text{open}}$ or $d_{\text{close}}$, but can see the LLM's response to their attack. In the game, this is depicted as in Fig. 2.
>
> Please let us know if we can make this more clear!
>
> ## Weakness 2
>
> > The paper should establish a connection to Textual Backdoor attacks, even though these attacks typically require a more significant level of access to LLMs or their pretraining data than the setting the authors are primarily interested in. This additional context would help improve the clear understanding of the threat model in which Tensor Trust operates.
>
> Thanks for the suggestion! We added the following paragraph to the related work section (section 7) to connect our work with past work on training-time attacks, including textual backdoor attacks. In this paragraph, [Dai et al.](https://arxiv.org/abs/1905.12457), [Chen et al.](https://arxiv.org/abs/2006.01043), [Qi et al.](https://arxiv.org/abs/2105.12400), and [Wallace et al.](https://arxiv.org/pdf/2010.12563.pdf) belong to this category. If we missed any work that you think would be useful to add to the paper, we would be happy to add them.
>
> > Other past work considers training-time attacks. This might include poisoning a model’s training set with samples that cause it to misclassify certain inputs at test time (Biggio et al., 2012; Dai et al., 2019; Chen et al., 2021; Qi et al., 2021; Wallace et al., 2020), or fine-tuning an LLM to remove safety features (Qi et al., 2023). These papers all assume that the attacker has some degree of control over the training process (e.g. the ability to corrupt a small fraction of the training set). In contrast, we consider only test-time attacks on LLMs that have already been trained.
>
> Please consider updating your score if these changes have adequately addressed your concerns about the paper, and let us know if there’s any other way that we can improve the writing.

---

> ### Comment · Reviewer_gj2Z · 2023-11-21
> **Reply**
>
> Thanks for the modifications. I have increased my score.

---

### Author Response · Authors · 2023-11-17
**Common response to all reviewers and AC**

Thank you to all reviewers for their helpful feedback! We have uploaded a substantially improved copy of the paper (as described below) and responded to each reviewer individually. Here's our high-level summary of the comments raised by reviewers and our responses:

* ​​Reviewer gj2Z commented that "use of an online game ... is a novel and creative way to understand the weaknesses of LLMs" an that the "benchmarks ... are a valuable contribution ... [as] a standardized way to assess the security of these models".
   However, they also commented that "the paper's setting is initially hard to grasp" (due to lack of notation & clear threat model) and that "the paper should establish a connection to textual backdoor attacks". In response, we:
	1. Addressed the first weakness by modifying section 2 to use precise notation in defining attacker and defender objectives and capabilities (i.e. the threat model)
	2. Addressed the second weakness by adding a paragraph on training-time attacks (including textual backdoor attacks) to the related work.
* Reviewer ZAuw noted that the dataset will be released publicly and that "samples in the dataset are high-quality since they are devised manually". In terms of weaknesses, they said that "the mechanism of this game is monotonous" and that they are "unsure whether the topic of this paper aligns with the theme of this conference". They also asked about potential applications and "whether the attack and defense in the dataset are still effective if the prompt of the opening defense is changed". Here's how we responded:
	1. To address the first weakness, we clarified that players seem to have enjoyed the game (as of today we have >330k attacks from >3,000 registered accounts) and that our simple mechanism has been sufficient to elicit diverse attack/defense strategies (including the 16 most common categories listed in Table 1).
	2. For the second weakness, we pointed out that ICLR has a history of publishing strong papers on benchmarks/datasets and robustness, and that this work is within the scope of the call for papers.
	3. In response to the first question, we improved the paper to highlight applications of the data that enhance LLM robustness.
	4. We answered the second question with some remarks on the relative importance of opening defenses.
* Reviewer 6PUw gave a positive review saying that the game "is well-designed and has the potential to serve as a benchmark for evaluating the adversarial robustness of Large Language Models". For weaknesses, they that the game "is not very suitable for collecting the prompt extraction dataset". We clarified that although the mechanism of our game does not have prompt extraction as the ultimate goal, players nevertheless found prompt extraction useful to win at the game, and so we have many prompt extraction attacks in our dataset. In particular, something like 27% of successful attacks in our original dataset used the defender's exact access code, indicating that they were the result of successful prompt extraction.

We've uploaded a new copy of the paper with these fixes. The paper also has a few improvements that were not requested by reviewers, but which we thought would be valuable for the community:


* We manually filtered the two benchmark subsets from section 3 so that every sample has had a human confirm that it is a valid prompt hijacking or extraction, respectively. Previously we were only using automated heuristics for filtering, which occasionally resulted in invalid samples (e.g. a very weak defense that would say "access granted" in response to any input). We've also expanded the benchmark subsets to 775 samples for the hijacking benchmark and 569 samples for the extraction benchmark. The experimental results have been updated accordingly, although we did not see much change in the rank order of LLMs as a result of these adjustments.
* We added an extra section to the appendix on prompt extraction detection, which is the task of inferring whether a given LLM output leaks enough information about the access code for the player to reconstruct the access code. This is a non-trivial task because the access code is sometimes leaked indirectly, like through hints. This small additional dataset may be useful for creating few-shot classifiers for prompt injection.
* We improved overall writing/presentation quality and fixed some typos.

Finally, we would like to note that our online game has continued to receive submissions, and we now have **more than 2.6× as much raw data** in our database as we did for the original submission (previously we had 126,000 submitted attacks, now over 330,000). We haven't updated the numbers in the paper yet because data is still rolling in, although we plan to release all the data we're getting (minus ToS-violating submissions) for the final version of the paper.

---

### Meta-Review · Area_Chair_sL2p · 2023-12-05

**Metareview:**

The paper presents a novel and valuable contribution to the field of Large Language Models through the development of Trust Tensor, an online game designed to collect human-generated adversarial examples for instruction-following LLMs. This innovative approach not only aids in understanding the weaknesses of these models but also in creating benchmarks for evaluating LLM resistance to prompt injection attacks, a crucial aspect in the development of secure LLM-based applications. The public release of the high-quality dataset generated from this game further enhances the paper's contribution. However, there are areas for improvement. The paper could benefit from a clearer explanation of the threat model and a more direct connection to textual backdoor attacks to enhance readers' understanding. While there are concerns about the monotony of the game mechanism and the relevance of the paper's topic to the conference theme, these do not significantly detract from the overall value of the work. The distinction between the task of prompt extraction and the Trust Tensor game is noted, but the game's potential as a benchmark for adversarial robustness is a compelling strength. Overall, the paper is recommended for acceptance due to its originality, practical importance, and contribution to the security of LLMs.

**Justification For Why Not Higher Score:**

N/A

**Justification For Why Not Lower Score:**

The complexity of the adversarial example collection mechanism, the intricacies of the threat model, and the detailed nature of the benchmark creation are aspects that may not be effectively communicated in the limited scope of a poster. Posters typically require concise presentations, and the depth of this research, particularly in terms of its technical and security-related aspects, necessitates a more comprehensive format for presentation. A full-length presentation would better accommodate the detailed explanations and discussions needed to fully convey the significance and technicalities of this work.

---

### Decision · Program_Chairs · 2024-01-16

Accept (spotlight)